

# The elusive 8.2 ka event in speleothems from southern France

Maddalena Passelergue[1,2], Isabelle Couchoud[1,2], Russell N. Drysdale[2], John Hellstrom[2], Dirk L. Hoffmann[3], Alan Greig[2]

[1]Laboratoire EDYTEM, UMR 5204 Université Savoie Mont Blanc-CNRS, *73376 Le Bourget du Lac*, France
[2]School of Geography, Earth and Atmospheric Sciences (SGEAS), University of Melbourne, *Parkville 3010 VIC*, Australia
[3]Georg-August-Universität Göttingen, GZG - Geochemistry and Isotope Geology, *37077 Göttingen*, Germany

*Correspondence to*: Maddalena Passelergue (mpasselergue@student.unimelb.edu.au)

**Abstract.** The Holocene is generally considered a climatically stable period, yet a prominent perturbation occurred around 8.2 ka BP. Evidence of its impacts has been identified in many palaeoclimate archives across Europe. However, outside the Atlantic seaboard, no clear high-resolution signal for this event has emerged from southwestern Europe. Here, we investigate the potential impact of the 8.2 ka event in southern France through high-resolution multiproxy analyses of two speleothems from the Ardèche region. Variations in Mg/Ca and Sr/Ca of the speleothem calcite are attributed to the prior calcite precipitation (PCP) effect and indicate switches between drier and wetter conditions. The $\delta^{13}C$ signal is likely influenced by soil development and biological activity, integrating both regional climate conditions and local geomorphology. The pattern of speleothem $\delta^{18}O$ changes do not correlate with regional palaeotemperature reconstructions and is therefore more likely related to hydrology, such as variations in the seasonality of karst recharge and/or the moisture source. During the 8.2 ka event, no distinct geochemical anomaly is recorded by the Ardèche speleothems, suggesting either a limited climatic impact in southern France or a lack of sensitivity of these speleothem proxies to an event of this magnitude. While the muted $\delta^{18}O$ response may be explained by a decoupling from temperature and buffering by Mediterranean influences at the time, the absence of a clear hydrological response in Mg/Ca, Sr/Ca and $\delta^{13}C$ remains unresolved. Therefore, despite a likely southward displacement of the westerlies induced by the 8.2 ka event, the Ardèche region may have remained under their influence, preventing a marked shift towards drier conditions. Consistent with other records from southern France, our results showing no significant changes around 8.2 ka challenge the spatial extent and uniformity of its climatic impacts across western Europe.

## 1 Introduction

The transition between the last glacial period and the Holocene is marked by significant climate and environmental changes, including increasing temperatures and greenhouse gases, ice-sheet melting and sea-level rise, and alteration of atmospheric and ocean circulation patterns (Bradley, 2005; Shi et al., 2020; Smith et al., 2011). Although the Holocene is characterised by a more stable regime, a series of major climate anomalies occurred (Bradley, 2005; Shi et al., 2020; Smith et al., 2011), the most prominent of which is the ~8.2 ka event (where ka is thousands of year before 1950 CE, as with all dates mentioned hereafter) (Alley et al., 1997; Rohling and Pälike, 2005; Thomas et al., 2007). A widely accepted trigger for the 8.2 ka event is a substantial release of freshwater from proglacial Lake Agassiz-Ojibway into the Hudson Bay during the late stages of the Laurentide ice sheet retreat (Barber et al., 1999). This massive freshwater discharge is believed to have altered the density and temperature of the ocean surface of the North Atlantic. The consequent slowing of the Atlantic Meridional Overturning Circulation (AMOC) and displacement of the deep-water convection zone (Barber et al., 1999; Ellison et al., 2006; Shi et al., 2020) resulted in a decrease in poleward heat transport and a cooling of Greenland surface air temperatures. This cooling, as well as the freshening of the ocean surface water recorded by lower $\delta^{18}O$ and $\delta^{15}N$ values in Greenland ice-cores, ultimately constrain the timing and duration of the 8.2 ka event to between 8.25±0.05 and 8.09±0.05 ka (Kobashi et al., 2007; North Greenland Ice Core Project members, 2004; Thomas et al., 2007). Due to its brief duration (~160 years; Thomas et al., 2007), studying the impacts of the 8.2 ka event requires well-dated, high-resolution palaeoenvironmental records.





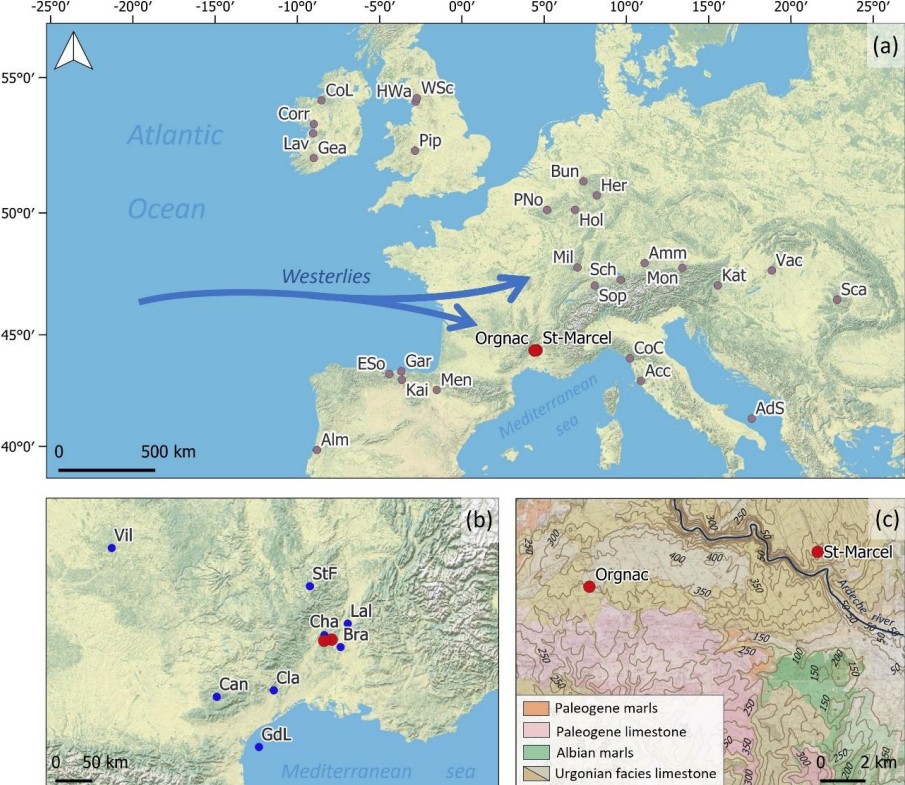


**Figure 1. (a)** Location of Aven d'Orgnac and St-Marcel Cave (red dots; this study) and other European sites mentioned in the text (grey dots). **(b)** Regional location of the two caves in this study (red dots) and other southern France paleo-records cited in the text (blue dots). **(c)** Geological and topographical settings (modified after the Bureau de Recherches Géologiques et Minières) of the study sites (red dots). *Acc*: Accesa lake (Magny et al., 2007); *AdS*: Adriatic Sea – core MD90-917 (Siani et al., 2013); *Alm*: Almonda cave (Benson et al., 2021); *Amm*:

Ammersee lake (Von Grafenstein et al., 1999); *Bra*: Mondragon-Les Brassières (Delhon, 2005); *Bun*: Bunker Cave (Waltgenbach et al., 2020); *Can*: Canroute peat (d'Oliveira et al., 2023); *Cha*: Chauvet Cave (Genty et al., 2006); *Cla*: Clamouse Cave (McDermott et al., 1999); *CoC*: Corchia Cave (Zanchetta et al., 2007) *CoL*: Cooney Lough (Ghilardi and O'Connell, 2013); *Corr*: Lough Corrib (Holmes et al., 2016); *ESo*: El Soplao Cave (Kilhavn et al., 2022); *Gar*: La Garma Cave (Baldini et al., 2019); *GdL*: Lion gulf marine cores (Jalali et al., 2016); *Gea*: Loch Gealain (Holmes et al., 2016); *Her*: Herbslabyrinth Cave (Waltgenbach et al., 2020); *Hol*: Holzmaar lake (Prasad et al., 2009);

*HWa*: Hawes Water (Marshall et al., 2007); *Kai*: Kaite Cave (Dominguez-Villar et al., 2009); *Kat*: Katerloch Cave (Boch et al., 2009); *Lal* : Espeluche-Lalo archeological site (Berger et al., 2016); *Lav* : Loch Avolla (Holmes et al., 2016); *Men*: Mendukilo Cave (Bernal-Wormull et al. 2023); *Mil*: Milandre Cave (Affolter et al., 2019); *Mon*: Mondsee lake (Schubert et al., 2023); *Pip*: Pippikin Cave (Daley et al., 2011); *PNo*: Père Noël cave (Allan et al., 2018); *Sca*: Scarisoara Cave (Perşoiu et al., 2017); *Sch*: Schleninsee lake (Tinner and Lotter, 2001); *Sop*: Soppensee lake (Tinner et Lotter, 2001); *StF* : Saint Front lake (Martin et al., 2020); *Vac*: Vacska Cave (Demény et al., 2023); *Vil*: Villars

Cave (Genty et al., 2006); *WSc*: White Scar Cave (Daley et al., 2011 ).

Europe is strongly impacted by conditions in the North Atlantic, particularly due to the influence of the westerlies. Thus, the 8.2 ka event is clearly recorded on the Atlantic seaboard as a decrease in the $\delta^{18}O$ of stalagmites from the Iberian Peninsula (Baldini et al., 2019; Benson et al., 2021; Dominguez-Villar et al., 2009; Kilhavn et al., 2022) to as far north as the British Isles (Daley et al., 2011). Although less pronounced, similar signals are evident in stalagmite proxies (e.g. Allan et al., 2018;

Demény et al., 2023; Waltgenbach et al., 2020; Fig. 1a) and ice cores (e.g. Perşoiu et al., 2017) located north and east of the Alps. Other palaeoenvironmental archives (Ghilardi and O'Connell, 2013; Von Grafenstein et al., 1999; Holmes et al., 2016; Marshall et al., 2007; Prasad et al., 2009; Schubert et al., 2023; Tinner and Lotter, 2001) also record the influence of the 8.2



ka event in northern and continental Europe. However, some records suggest that the presence of sapropel S1, which occurred at the same time, masks the possible influence of the 8.2 ka event in some regions (e.g. Magny et al., 2007; Siani et al., 2013; Zanchetta et al., 2007). More generally, there are no high-resolution records in the western Mediterranean region showing significant changes that can be clearly attributed to the 8.2 ka event (e.g. d'Oliveira et al., 2023; Jalali et al., 2016; McDermott et al., 1999; Fig. 1a, b).

Despite its proximity to the Atlantic Ocean and the dominance of the mid-latitude westerlies, few high-resolution data are available for this period from France. Of these, only the δ¹⁸O from a Villars Cave speleothem appears to have recorded variations that could be related to the 8.2 ka event (unpub. thesis, J. Ruan, 2016; Fig. 1b). Further south, in the Rhône Valley, evidence from anthracological, geomorphological and phytolith data from Lalo and Les Brassières sites suggests the occurrence of more frequent, short and intense rainfall during the 8.2-8.1 ka period, synchronous with a high fire regime. These conditions, occurring during a period marked by an overall dry climate, led to soil degradation and erosion, and may be linked to the hydrological impact of the 8.2 ka event (Berger et al., 2016; Delhon, 2005). However, anthracological assemblages from both Lalo and Les Brassières do not support perturbations of sufficient intensity to alter vegetation dynamics at 8.2 ka (Delhon, 2005). The isotopic records from Chauvet (Genty et al., 2006) and Clamouse (McDermott et al., 1999) Caves do not show any clear response to the 8.2 ka event. Similarly, high-resolution sediment-core records from the Canroute peat bog (d'Oliveira et al., 2023) and the Gulf of Lion (Jalali et al., 2016) do not indicate any significant changes that could be associated with this event. Therefore, the nature of the impacts of the 8.2 ka event in France, and more broadly in southern Europe, needs further substantiation.

This study aims to contribute to the pool of data for this region by presenting new well-dated, high-resolution stalagmite records from southeastern France. We applied a multiproxy approach on two stalagmites collected from neighbouring caves, with a particular focus on the period 11.5 to 5.5 ka, in order to contextualise the 8.2 ka event and investigate its potential climatic impact in southeastern France.

## 2 Materials and Methods

### 2.1 Study sites

The studied stalagmite samples come from two caves located on the Ardèche limestone plateau, approximately 15 km west of the Rhône Valley (Fig. 1). The Upper Cretaceous limestone (Urgonian facies, very pure reef limestone) plateau is incised by the Ardèche River, creating a gorge that meanders down to Pont Saint-Esprit. The surface of the plateau displays extensive karren and other dissolution features, and is characterised by thin soil or bare karst, with pockets of soil trapped in the surface dissolution features. A typical Mediterranean garrigue vegetation (low, thermophilic, and dominated by *Quercus ilex*) is present on the plateau. Saint-Marcel Cave is a several-km-long network located in the Ardèche gorge (natural entrance: 44°19'37" N, 4°32'20" E; Fig. 1c), extending under a watershed lying between ~280 and 80 m a.s.l. The Aven d'Orgnac, situated 10 km further west, has a natural sinkhole entrance open on the plateau at 312 m a.s.l (44°19'11" N, 4°24'43" E; Fig. 1c). The proximity to the Mediterranean Sea (about 100 km to the south) results in a mixed meteorological influence, with rain-bearing air masses coming from both the Atlantic and the Mediterranean (Celle-Jeanton et al., 2001). More specifically, a two-year daily precipitation study at Avignon (~50 km south of Orgnac and St Marcel caves) indicates that, despite a similar proportion of rain events from the Atlantic, the Mediterranean or of mixed origin (i.e. mid-latitude Atlantic Ocean, Western Mediterranean and Iberian peninsula origin), 52% of the total precipitation is of Mediterranean origin, 12% from the North Atlantic and the rest of mixed origin (Celle-Jeanton et al., 2001). In Ardèche, the meteorological station located at Orgnac (Genty et al., 2014) highlights a climate characterised by hot, dry summers followed by a significant increase in Mediterranean-origin precipitation



in autumn (the so-called "Cevenol episodes"), which contributes notably to the karst recharge. The annual mean air temperature at Orgnac is 14.1±0.5 °C, while the annual precipitation averages 934±328 mm (data collected between 2001 and 2011; Genty et al., 2014).

Drip-water monitoring has been carried out over 10 years at Chauvet Cave, near St-Marcel and Orgnac Caves (Genty et al., 2014). The main results indicate a significant inter-annual mixing of the rainwater (with the exception of August rainfall). Since the host rock and fracture network are similar at all sites, we can expect the same mixing of infiltration waters at St Marcel and Orgnac Caves. This implies that the δ¹⁸O of the calcite represents at least an intra-annual, if not an inter-annual, average of the δ¹⁸O of recharge precipitation.

**2.2 Speleothem samples**

SM1-A is a fragment of a larger broken stalagmite collected on the floor of a large gallery within Network 1 of Saint-Marcel Cave (site called the "Gothic chapel"; ~1400 m from the natural entrance; 120 m deep below the surface) in 2016. This stalagmite, whose base is missing, grew episodically since at least 123 ka. Only the upper 24 cm, formed during the Holocene, were investigated for this study. The growth period of this segment spans 11.3 to 0.2 ka.

Stalagmite OR09-A was collected in 2009 from the Aven d'Orgnac, in the chamber "Salle 1" of the network section called "Orgnac II" (~200 m from the natural entrance, a large sinkhole that likely opened during the Pleistocene; ~110 m deep below the surface). It was standing *in situ* as a single piece at the time of collection and grew on a mud embankment. It exhibits a growth period between 11.1 and 0.5 ka without any visible hiatus over its 44 cm length (Fig. A1).

Both stalagmites are composed of fairly pure calcite, except for the base of OR09-A, which shows episodes of clay
incorporation from rising groundwaters that submersed the stalagmite when the Ardeche River was in flood. However, these episodes gradually diminish towards the top until they disappear. The calcite fabric is compact columnar (Frisia et al., 2000), and lamination is very faint. OR09-A grew very regularly and straight, while SM1-A displays gentle growth axis shifts and several faint discontinuities, potentially associated with growth hiatuses (Fig. A2).

**2.3 U-Th dating**

Age models were constructed from 17 uranium-thorium (U-Th) dates for SM1-A and 23 dates for OR09-A (Tables 1 and 2; Fig. 2). Samples were extracted using a dental air drill, yielding prisms of calcite each weighing approximately 200 mg. With the exception of eight OR09-A dating samples that were chemically prepared following the method detailed in Hoffmann et al. (2016), all other OR09-A and SM1-A dating samples were prepared following the method of Hellstrom (2003). The U and Th isotopic ratios were measured using a MC-ICP-MS (a Thermo Finnigan *Neptune* at the National Centre for Research on
Human Evolution (CENIEH) in Burgos (Spain) following methods outlined in Hoffmann et al. (2007), and Nu Instruments *Plasma* at the University of Melbourne). An empirically derived initial thorium ($^{230}$Th/$^{232}$Th$_{initial}$) correction was applied to the ages, as described by Corrick et al. (2020), to reduce analytical error due to the presence of detrital thorium. An age-depth model for each stalagmite was built from these dates using the Finite Positive Growth Rate algorithm described by Corrick et al. (2020).




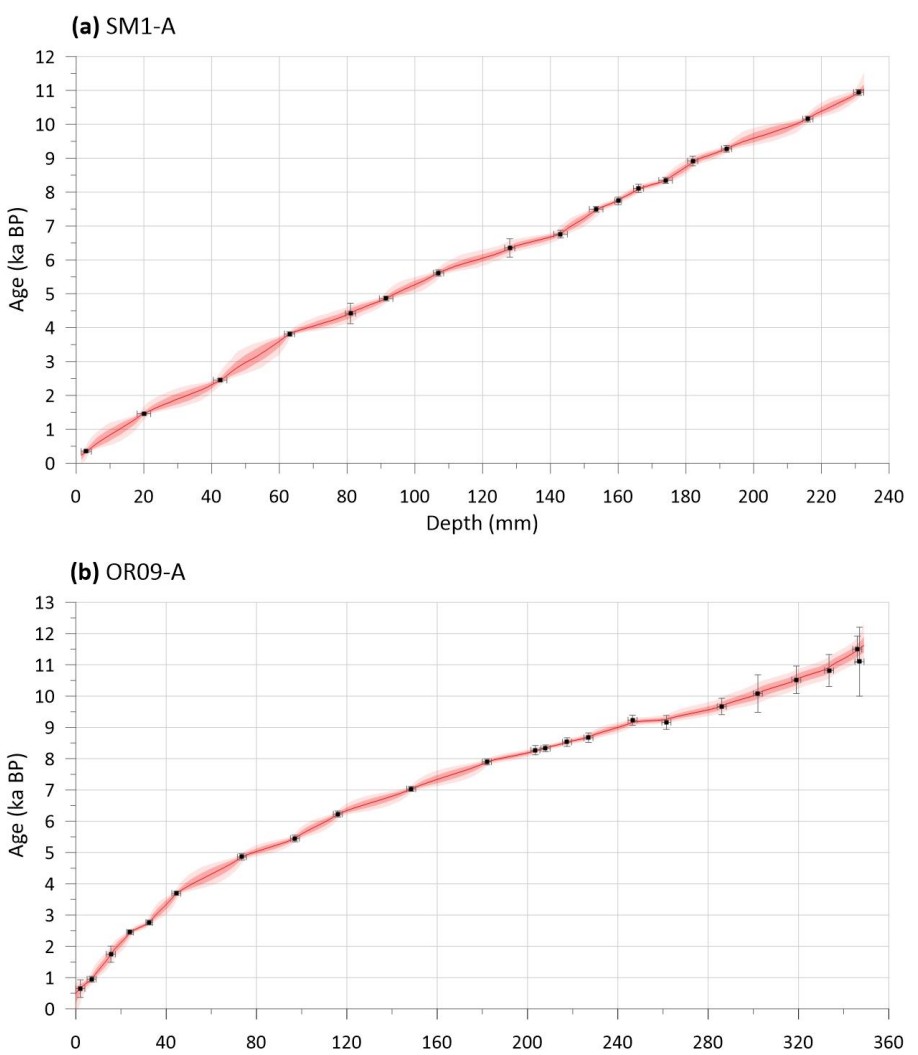

**Figure 2.** Age-depth models of **(a)** SM1-A (St-Marcel Cave, France) and **(b)** OR09-A (Aven d'Orgnac, France), calculated following the method described in Corrick et al. (2020). Dark and light red shaded areas correspond to 66% and 95% uncertainty envelopes. Depth is relative to the top of the stalagmite, and age is expressed in ka BP (1950).




**Table 1** Uranium series analysis on stalagmite SM1-A determined by Hellstrom (2006) procedure. The analytical errors are provided with a 95% uncertainty. Age in ka before 1950 AD corrected for initial $^{230}$Th using eqn. 1 of Hellstrom (2006), the decay constants of Cheng et al. (2013) and [$^{230}$Th/$^{232}$Th]i of 0.48 ± 0.32.

| Sample ID | Depth (mm) | Mass (g) | $^{238}$U (ng.g$^{-1}$) | $^{230}$Th/$^{238}$U ×10$^{-3}$ activity ratio | $^{234}$U/$^{238}$U ×10$^{-3}$ activity ratio | $^{232}$Th/$^{238}$U ×10$^{-4}$ activity ratio | $^{230}$Th/$^{232}$Th activity ratio | Age (ka) uncorrected | Age ka BP (1950) corrected | $^{234}$U/$^{238}$U ×10$^{-3}$ initial activity ratio |
|---|---|---|---|---|---|---|---|---|---|---|
| SM1-11 | 3 ± 1.5 | 0.20 | 154 ± 31 | 3.2 ± 0.2 | 820.2 ± 1.9 | 0.97 ± 0.02 | 35.1 | 0.426 ± 0.027 | 0.348 ± 0.027 | 820.0 ± 1.9 |
| SM1-4 | 20 ± 2.0 | 0.40 | 149 ± 11 | 11.4 ± 0.2 | 815.9 ± 1.5 | 0.67 ± 0.01 | 168.7 | 1.535 ± 0.027 | 1.458 ± 0.028 | 815.1 ± 1.5 |
| SM1-15 | 42.5 ± 2.0 | 0.20 | 161 ± 32 | 18.1 ± 0.3 | 791.4 ± 2.5 | 0.14 ± 0.01 | 1323.3 | 2.525 ± 0.043 | 2.452 ± 0.043 | 789.9 ± 2.5 |
| SM1-7 | 63 ± 1.5 | 0.14 | 132 ± 10 | 26.5 ± 0.4 | 756.8 ± 2.1 | 0.68 ± 0.01 | 391.2 | 3.894 ± 0.061 | 3.817 ± 0.062 | 754.1 ± 2.1 |
| SM1-10 | 81 ± 1.5 | 0.13 | 116 ± 23 | 30.3 ± 2.0 | 750.8 ± 1.4 | 0.96 ± 0.28 | 337.6 | 4.499 ± 0.304 | 4.42 ± 0.30 | 747.6 ± 1.4 |
| SM1-16 | 91.5 ± 2.0 | 0.19 | 113 ± 23 | 33.2 ± 0.4 | 752.7 ± 2.5 | 0.16 ± 0.01 | 2059.4 | 4.930 ± 0.063 | 4.857 ± 0.064 | 749.2 ± 2.6 |
| SM1-8 | 107 ± 1.5 | 0.16 | 121 ± 24 | 38.4 ± 0.6 | 757.4 ± 1.6 | 0.41 ± 0.01 | 944.3 | 5.682 ± 0.090 | 5.614 ± 0.094 | 753.5 ± 1.7 |
| SM1-9 | 128 ± 1.5 | 0.14 | 106 ± 21 | 42.5 ± 1.7 | 745.3 ± 1.9 | 0.90 ± 0.16 | 492.6 | 6.425 ± 0.267 | 6.35 ± 0.27 | 740.6 ± 2.0 |
| SM1-17 | 143 ± 2.0 | 0.17 | 108 ± 22 | 45.0 ± 0.7 | 743.7 ± 2.3 | 0.27 ± 0.01 | 1657.3 | 6.829 ± 0.112 | 6.76 ± 0.11 | 738.7 ± 2.4 |
| SM1-3 | 153.5 ± 2.0 | 0.39 | 117 ± 9 | 49.0 ± 0.5 | 733.3 ± 1.3 | 0.63 ± 0.01 | 783.2 | 7.572 ± 0.081 | 7.495 ± 0.081 | 727.5 ± 1.3 |
| SM1-2 | 160 ± 1.0 | 0.15 | 129 ± 10 | 50.9 ± 0.7 | 738.4 ± 1.4 | 0.80 ± 0.03 | 640.2 | 7.820 ± 0.113 | 7.74 ± 0.11 | 732.6 ± 1.4 |
| SM1-12 | 166 ± 1.5 | 0.15 | 120 ± 24 | 53.2 ± 0.7 | 739.1 ± 2.1 | 0.36 ± 0.01 | 1492.1 | 8.180 ± 0.113 | 8.11 ± 0.12 | 733.0 ± 2.2 |
| SM1-6 | 174 ± 2.0 | 0.11 | 109 ± 8 | 54.6 ± 0.5 | 737.5 ± 2.2 | 0.82 ± 0.02 | 666.8 | 8.424 ± 0.085 | 8.346 ± 0.085 | 731.2 ± 2.3 |
| SM1-13 | 182 ± 1.5 | 0.16 | 80 ± 16 | 59.2 ± 0.9 | 750.4 ± 2.1 | 1.06 ± 0.02 | 562.5 | 9.000 ± 0.147 | 8.92 ± 0.14 | 744.0 ± 2.2 |
| SM1-14 | 192 ± 1.5 | 0.11 | 111 ± 22 | 60.9 ± 0.6 | 744.4 ± 2.1 | 0.22 ± 0.01 | 2800.4 | 9.352 ± 0.100 | 9.28 ± 0.10 | 737.6 ± 2.2 |
| SM1-5 | 216 ± 1.5 | 0.10 | 110 ± 8 | 69.2 ± 0.5 | 775.5 ± 2.2 | 1.34 ± 0.03 | 516.3 | 10.237 ± 0.084 | 10.155 ± 0.086 | 768.9 ± 2.3 |
| SM1-1 | 231 ± 1.5 | 0.20 | 107 ± 8 | 8.22 ± 0.6 | 0.8572 ± 1.5 | 1.08 ± 0.03 | 760.1 | 11.023 ± 0.088 | 10.943 ± 0.087 | 852.7 ± 1.5 |

**Table 2** Uranium series analysis on stalagmite OR09-A determined by Hellstrom (2006; dates marked with an asterisk) or Hoffmann et al. (2007) / Hoffmann et al. (2016) procedure. The analytical errors are provided with a 95% uncertainty. Age in ka before 1950 AD corrected for initial $^{230}$Th using eqn. 1 of Hellstrom (2006), the decay constants of Cheng et al. (2013) and [$^{230}$Th/$^{232}$Th]i of 0.48 ± 0.24.

| Sample ID | Depth (mm) | Mass (g) | $^{238}$U (ng.g$^{-1}$) | $^{230}$Th/$^{238}$U ×10$^{-3}$ activity ratio | $^{234}$U/$^{238}$U ×10$^{-3}$ activity ratio | $^{232}$Th/$^{238}$U ×10$^{-4}$ activity ratio | $^{230}$Th/$^{232}$Th activity ratio | Age (ka) uncorrected | Age ka BP (1950) corrected | $^{234}$U/$^{238}$U ×10$^{-3}$ initial activity ratio |
|---|---|---|---|---|---|---|---|---|---|---|
| Or-S1-09-A #8 | 2.0 ± 2.0 | 0.17 | 50 ± 0.2 | 9.4 ± 0.4 | 810.9 ± 1.8 | 86.73 ± 0.72 | 1.1 | 1.268 ± 0.054 | 0.65 ± 0.28 | 810.9 ± 1.8 |
| OR-A-18* | 7.0 ± 2.0 | 0.20 | 42 ± 8 | 8.2 ± 0.6 | 813.7 ± 2.4 | 14.41 ± 0.67 | 6.0 | 1.104 ± 0.082 | 0.95 ± 0.09 | 813.7 ± 2.4 |
| Or-S1-A/01 | 15.5 ± 2.0 | 0.06 | 65 ± 0.4 | 16.0 ± 1.4 | 821.4 ± 3.2 | 53.90 ± 0.41 | 3.0 | 2.153 ± 0.191 | 1.75 ± 0.26 | 821.4 ± 3.2 |
| OR-A-9* | 24.0 ± 1.5 | 0.19 | 69 ± 14 | 19.3 ± 0.4 | 824.5 ± 1.7 | 10.96 ± 0.22 | 17.6 | 2.581 ± 0.061 | 2.45 ± 0.07 | 824.5 ± 1.7 |
| OR-A-19* | 32.5 ± 1.5 | 0.17 | 59 ± 12 | 21.7 ± 0.5 | 824.5 ± 2.3 | 11.63 ± 0.38 | 18.9 | 2.897 ± 0.069 | 2.76 ± 0.08 | 828.5 ± 2.3 |
| Or-S1-09-A #7 | 44.5 ± 2.0 | 0.20 | 77 ± 0.3 | 28.5 ± 0.4 | 831.0 ± 1.5 | 8.06 ± 0.07 | 35.3 | 3.813 ± 0.055 | 3.70 ± 0.06 | 831.0 ± 1.5 |
| OR-A-10* | 73.5 ± 2.0 | 0.20 | 70 ± 14 | 37.5 ± 0.7 | 844.8 ± 2.0 | 6.76 ± 0.14 | 55.5 | 4.958 ± 0.101 | 4.86 ± 0.10 | 844.8 ± 2.0 |
| OR-A-20* | 97.0 ± 2.0 | 0.19 | 63 ± 13 | 42.1 ± 0.8 | 847.7 ± 2.8 | 8.67 ± 0.17 | 48.9 | 5.562 ± 0.110 | 5.45 ± 0.11 | 847.7 ± 2.8 |
| Or-S1-09-A #6 | 116.0 ± 2.0 | 0.17 | 71 ± 0.3 | 47.9 ± 0.7 | 850.9 ± 1.7 | 7.87 ± 0.09 | 60.9 | 6.343 ± 0.100 | 6.23 ± 0.10 | 850.9 ± 1.7 |
| OR-A-13* | 148.5 ± 2.0 | 0.23 | 78 ± 16 | 53.7 ± 0.5 | 849.7 ± 2.1 | 6.05 ± 0.13 | 89.2 | 7.132 ± 0.071 | 7.03 ± 0.08 | 849.7 ± 2.1 |
| Or-S1-09-A #5 | 182.0 ± 2.0 | 0.26 | 68 ± 0.3 | 59.7 ± 0.6 | 842.7 ± 1.4 | 11.44 ± 0.15 | 52.1 | 8.040 ± 0.090 | 7.90 ± 0.10 | 842.7 ± 1.4 |
| OR-A-11* | 203.5 ± 2.0 | 0.20 | 68 ± 14 | 62.2 ± 0.9 | 834.8 ± 1.9 | 22.40 ± 0.45 | 27.8 | 8.467 ± 0.129 | 8.27 ± 0.15 | 834.8 ± 1.9 |
| OR-A-14* | 208.0 ± 2.0 | 0.22 | 66 ± 13 | 61.9 ± 0.7 | 835.2 ± 2.3 | 4.29 ± 0.09 | 145.3 | 8.418 ± 0.102 | 8.33 ± 0.11 | 835.2 ± 2.3 |
| OR-A-15* | 217.5 ± 2.0 | 0.16 | 64 ± 13 | 63.5 ± 0.10 | 836.1 ± 2.2 | 8.34 ± 0.20 | 76.9 | 8.634 ± 0.144 | 8.53 ± 0.14 | 836.1 ± 2.2 |
| Or-S1-09-A #4 | 227.0 ± 2.0 | 0.13 | 74 ± 0.3 | 64.7 ± 0.8 | 827.6 ± 1.9 | 26.59 ± 0.21 | 24.3 | 8.910 ± 0.120 | 8.67 ± 0.15 | 827.6 ± 1.9 |
| OR-A-21* | 246.5 ± 2.0 | 0.17 | 63 ± 13 | 67.9 ± 0.10 | 823.6 ± 2.5 | 19.20 ± 0.32 | 35.6 | 9.410 ± 0.149 | 9.23 ± 0.16 | 823.6 ± 2.5 |
| OR-A-16* | 261.5 ± 2.0 | 0.15 | 72 ± 14 | 69.1 ± 0.8 | 822.6 ± 1.8 | 57.88 ± 1.10 | 12.0 | 9.596 ± 0.120 | 9.16 ± 0.22 | 822.6 ± 1.8 |
| Or-S1-A/02 | 286.0 ± 2.0 | 0.16 | 126 ± 0.6 | 72.5 ± 0.7 | 814.7 ± 2.0 | 72.80 ± 0.21 | 10.0 | 10.215 ± 0.111 | 9.67 ± 0.26 | 814.7 ± 2.0 |
| OR-A-22* | 302.0 ± 2.0 | 0.19 | 67 ± 13 | 82.5 ± 0.7 | 819.4 ± 2.3 | 177.26 ± 3.89 | 4.5 | 11.288 ± 0.153 | 10.08 ± 0.60 | 819.4 ± 2.3 |
| OR-A-17* | 319.0 ± 2.0 | 0.21 | 83 ± 17 | 82.5 ± 0.9 | 832.2 ± 2.5 | 133.59 ± 2.38 | 6.2 | 11.424 ± 0.138 | 10.52 ± 0.44 | 832.2 ± 2.5 |
| OR-A-12bis* | 333.5 ± 2.0 | 0.21 | 94 ± 19 | 85.9 ± 0.7 | 836.7 ± 2.1 | 157.21 ± 3.61 | 5.5 | 11.853 ± 0.109 | 10.82 ± 0.51 | 836.7 ± 2.1 |
| OR-A-23* | 346.0 ± 2.0 | 0.18 | 94 ± 19 | 89.9 ± 0.6 | 841.2 ± 2.2 | 128.99 ± 2.57 | 7.0 | 12.367 ± 0.095 | 11.50 ± 0.42 | 841.2 ± 2.2 |
| Or-S1-09-A #3 | 348 ± 3.0 | 0.16 | 87 ± 0.3 | 96.4 ± 0.10 | 839.8 ± 1.7 | 341.89 ± 0.78 | 2.8 | 13.360 ± 0.146 | 11.1 ± 1.1 | 839.8 ± 1.7 |



### 2.4 Stable isotopes of carbon and oxygen

To obtain the variation over time of $\delta^{18}O$ and $\delta^{13}C$, calcite powder samples were taken along the growth axis of the stalagmites. The samples were collected in powder form (~1 mg) using a CNC Micromilling lathe equipped with a 1 mm drill bit, and moving with a 1 mm step along the growth axis. The samples were analysed at the University of Melbourne on a continuous-flow IRMS (Analytical Precision *AP2003*) after digestion in 105% orthophosphoric acid. Normalisation to the VPDB scale was carried out using two standards (NEW1 and NEW12) previously calibrated against the standard international reference materials, NBS18 and NBS19. The $1\sigma$ analytical uncertainty on the sample measurements is 0.05 ‰ for $\delta^{13}C$ and 0.1 ‰ for $\delta^{18}O$.

### 2.5 Trace Elements

The analysis of magnesium (Mg) and strontium (Sr) concentrations relative to calcium (Ca) was carried out for SM1-A on subsamples of the same calcite powders used for isotopic analyses (i.e. at 1 mm step), making them directly comparable. Approximately 0.3 to 0.5 mg of calcite was dissolved in 15 mL of 2.5% twice-distilled $HNO_3$. Measurements were obtained using an Agilent 7700x quadrupole ICP-MS at the University of Melbourne. A mixed solution of sample digests was used to determine appropriate concentrations for the calibration standards and this mixture was also analysed every 10 samples to correct for instrument drift. The ratio of trace elements relative to $^{43}Ca$ are based on 10 replicates of 500 measurements for each sample. Standards were run as unknowns every 19 samples to estimate the reproducibility, which presents a Relative Standard Deviation (RSD) of 0.13 and 0.40 % for Mg/Ca and Sr/Ca respectively.

## 3 Results

### 3.1 Dating, age model, and growth rates

All ages from the two stalagmites are in stratigraphic order and are expressed in ka BP (1950). The concentration of $^{238}U$ ranges from 80 to 161 ng g$^{-1}$ for SM1-A and from 42 to 126 ng g$^{-1}$ for OR09-A. SM1-A consists mostly of relatively clean calcite (high $^{230}Th/^{232}U$ activity ratios; Table 1). In contrast, the $^{230}Th/^{232}U$ activity ratios for OR09-A are generally below 100, even below 10 in some cases, indicating significant detrital contamination (Table 2). The empirically derived ($^{230}Th/^{232}Th_{initial}$) corrections applied were 0.48±0.32 for SM1-A and 0.48±0.24 for OR09-A.

Both SM1-A and OR09-A display fairly constant growth rates between the early Holocene and 5.5 ka; the faint discontinuities observed in SM1-A cannot be resolved by the age-depth model, but nonetheless short growth hiatuses cannot be excluded just before ~8.4 ka and between ~7.7 and 7.5 ka (Fig. A2). The growth rate of SM1-A varies between 13 and 30 μm yr$^{-1}$, while OR09-A grew about twice faster, varying between 28 and 67 μm yr$^{-1}$. Nevertheless, it can be noted that growth rate was rather slow and stable for both stalagmites.

### 3.2 Isotopic Equilibrium Conditions

The Hendy test (Hendy, 1971) was not performed on SM1-A and OR09-A since it has been demonstrated that Hendy test criteria are not always reliable, and suitable sampling is difficult to perform (Dorale and Liu, 2009). However, the absence of consistent covariation between $\delta^{13}C$ and $\delta^{18}O$ in each of the stalagmites (Fig. 3) indicates that calcite precipitation was not associated with significant kinetic fractionation.



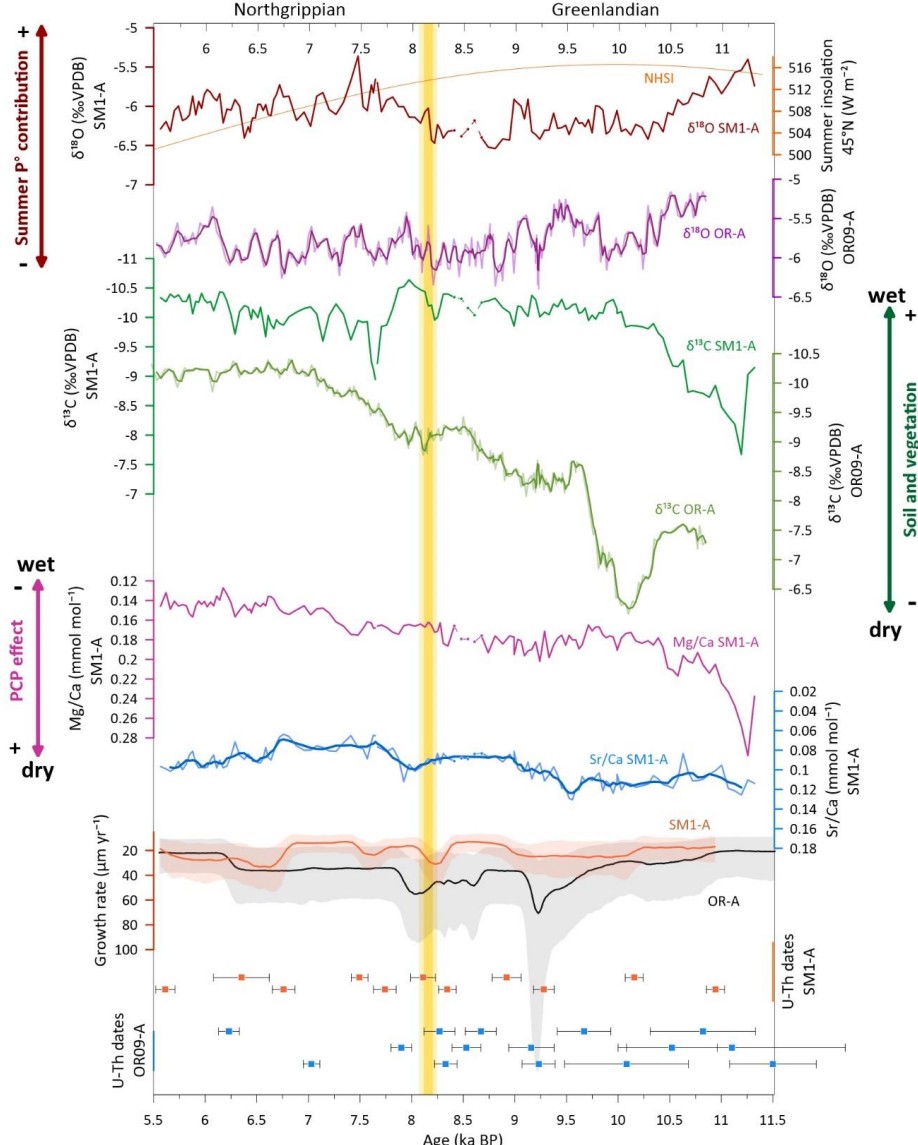

**Figure 3.** Palaeoclimate proxy data from SM1-A (St-Marcel Cave, southern France) and OR09-A (Aven d'Orgnac, southern France). From top to bottom: July insolation at 45°N (Laskar et al., 2004); δ¹⁸O of SM1-A and OR09-A; δ¹³C of SM1-A and OR09-A; Mg/Ca and Sr/Ca (thick line: running average with a window of five values) of SM1-A; growth rate of OR09-A and SM1-A; U-Th ages with their 2σ uncertainties for SM1-A and OR09-A. The shaded light-yellow bar marks the 8.2 event period (Thomas et al., 2007), and the darker part highlights the small variations questioned in SM1-A and OR09-A proxies. Dots and broken lines in SM1-A times series represent possible discontinuities not resolved by age-depth model. The δ¹⁸O and δ¹³C time-series of OR09-A have been smoothed (running average with a window of two values, thick lines) in order to display a similar temporal resolution to SM1-A time-series.

For most of the growth interval, the δ¹⁸O signals of these stalagmites show similar trends with a limited amplitude (Fig. 4a). This is expected because this isotope ratio is less affected by the nuances of drip-water catchment heterogeneities (e.g. vegetation cover, soil thickness, karst plumbing) than δ¹³C (and trace elements). Indeed, besides some discrepancy during the periods 9.7–9.3 ka and 8.0–7.6 ka, the two δ¹⁸O patterns are very similar. This is in spite of an offset for much of the period,





with OR09-A having generally higher values (Fig. 4a and b). This latter feature may be due to local edaphic factors, where infiltration waters may have been subjected to higher evapotranspiration. This explanation seems applicable for the period between 10.9 and 7.8 ka, where OR09-A $\delta^{18}$O is up to 1 ‰ higher; notably, the offset decreases with time until the two profiles converge at ~7.8 ka (Fig. 4a and b). There is a more pronounced offset between the two $\delta^{13}$C profiles across this same period.

The higher OR09-A $\delta^{13}$C values are consistent with edaphic/vegetation processes, whilst the timing of the convergence of the two $\delta^{13}$C profiles is almost identical to that for the $\delta^{18}$O profiles (Fig. 4c and d). For these reasons, we argue against significant in-cave kinetic fractionation and instead attribute anomalies between isotope profiles to differences in surface conditions above each drip site. We therefore affirm the robustness of the records as potential palaeoclimate indicators (Couchoud, 2008; Lachniet, 2009).

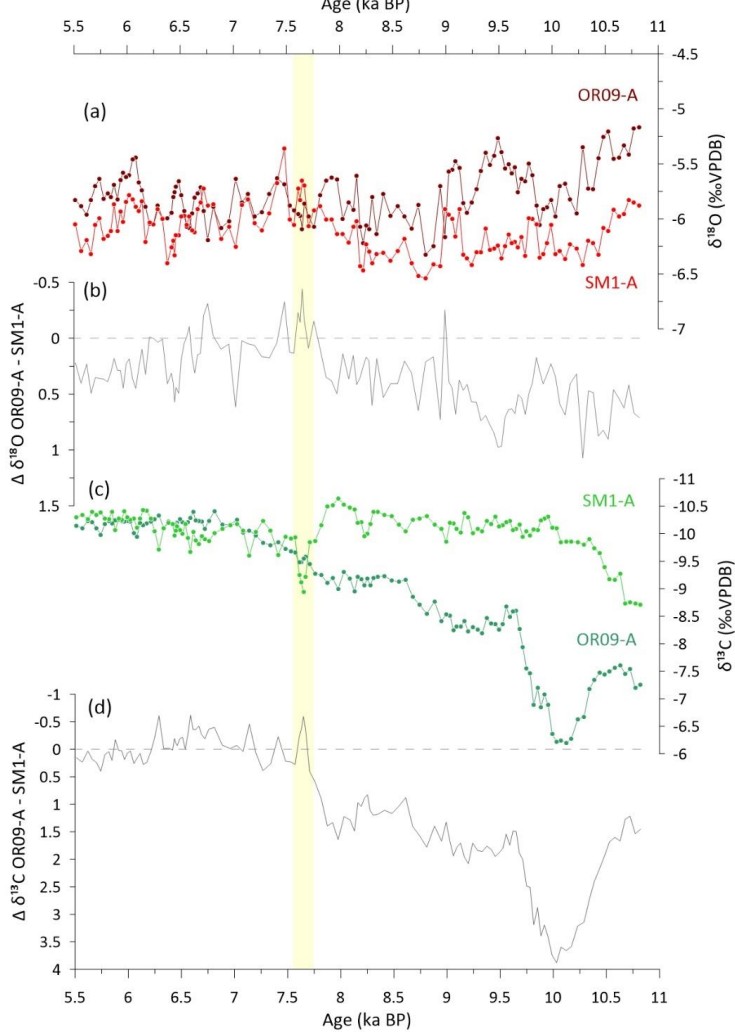

**Figure 4. (a) and (c)** SM1-A (St Marcel Cave, Ardèche; light red and green lines) and OR09-A (Aven d'Orgnac; dark red and green lines) $\delta^{18}$O and $\delta^{13}$C. OR09-A signal has been interpolated to fit with SM1-A age depth model. **(b) and (d)** The difference between both time series ($\Delta \delta^{18}$O and $\Delta \delta^{13}$C of OR09-A - SM1-A; grey solid lines). The dashed grey lines (0 value) are when the $\delta^{18}$O or $\delta^{13}$C of both stalagmites have no difference. The shaded yellow area indicates the timing of the convergence of the $\delta^{18}$O and $\delta^{13}$C profiles of both stalagmites




### 3.3 Variations in Stable Isotopes and Trace Elements

The δ¹⁸O of SM1-A calcite varies from -6.5 to -5.4 ‰, while that of OR09-A varies from -6.4 to -5.2 ‰ during the period 11.5–5.5 ka. As noted above, generally both δ¹⁸O signals follow similar trends: beyond secular variations, a decrease is observed during the Greenlandian, followed by stabilisation during the Northgrippian (Fig. 3 and 4).

The δ¹³C values of both SM1-A and OR09-A show a decreasing trend during the Greenlandian; however, major changes occur with a different timing (Fig. 3 and 4). Both records show, at first, a rapid increase of δ¹³C values up to reach a peak at ~11.2 ka in SM1-A and ~1100 years later in OR09-A. Then, the δ¹³C decreases sharply, until ~9.9 ka in SM1-A, and ~9.6 ka in OR09-A. Despite a peak event at 7.6±0.1 ka in SM1-A (Fig. 3), likely related to the growth discontinuity identified on the polished section (Fig. A2), the δ¹³C of SM1-A stabilises through the rest of the record at around -10.2 ‰. In contrast, the δ¹³C of OR09-

A shows first a gentle increase between ~9.6 and ~9.3 before a progressive decrease until ~7.1 ka, where it stabilises at around the same values as SM1 (Fig. 4c and d).

SM1-A magnesium concentration is generally low for speleothems, with Mg/Ca ranging from 0.13 to 0.30 mmol mol⁻¹, and displaying a decreasing trend over the period of interest. It notably shows a peak centred at ~11.2 ka, synchronous with the δ¹³C peak in the same stalagmite (Fig. 3). The Sr/Ca ratio ranges from 0.06 to 0.13 mmol mol⁻¹. The highest values are between

~11.3 and ~9.5 ka, after which there is a gradual decrease until ~6.7 ka, before a slight increase again until ~5.5 ka (Fig. 3).

Spearman $\rho$ correlation coefficients were calculated between each of the four SM1-A proxies over the entire considered period (i.e. 11.5 to 5.5 ka) as well as between the growth rate and the SM1-A proxies for the period 10.9 to 5.5 ka (i.e. the non-extrapolated period encompassed by the age-depth model). The coefficients have values equal to or below |0.46|, indicating weak to moderate correlation of the different proxies (Table 3).

**Table 3.** Spearman correlation coefficients ($\rho$) calculated among the different proxies of SM1-A. These tests were conducted on data from the period 11.5 – 5.5 Ka BP, except for correlation tests involving the growth rate, which is only available for the period 10.9 – 5.5 ka BP. Asterisks indicate correlation coefficients that are statistically significant at the 5% significance level (p < 0.05).

|  | δ¹³C | Mg/Ca | Sr/Ca | Growth rate |
|---|---|---|---|---|
| δ¹⁸O | 0.44* | -0.07 | -0.25* | 0 |
| δ¹³C | - | 0.37* | -0.03 | -0.13 |
| Mg/Ca |  | - | 0.46* | -0.46* |
| Sr/Ca |  |  | - | 0.06 |

In the time range corresponding to the 8.2 ka event defined by Thomas et al. (2007), i.e. between 8.25±0.05 and 8.09±0.05 ka,

only a slight increase of δ¹⁸O is identified between ~8.2 and ~8.1 ka in both SM1-A and OR09-A, synchronous with a slight δ¹³C increase in OR09-A. However, these variations do not exceed the signal amplitude of the previous 1000-year period (9.3–8.3 ka; Fig. 3), and the other proxies show no significant excursions. Thus, the 8.2 ka event is not clearly identifiable in the proxy series of these speleothems.




## 4 Discussion

In the following sections, our interpretation of the geochemical proxies for SM1-A and OR09-A speleothems described above will be developed, before focusing on the 8.2 ka event period and then discussing the climatic implications of our results in southeastern France.

### 4.1 Interpretation of proxies

The $\delta^{13}C$ of speleothem calcite can vary based on several, often competing, factors, including vegetation type, soil biogenic activity, and prior calcite precipitation (PCP), all of which are linked to the hydrological regime (McDermott, 2004). Considering that C3 vegetation taxa predominantly prevailed in the south of France during the Holocene (e.g. Berger et al., 2016; Martin et al., 2020), it is unlikely that the observed $\delta^{13}C$ variations were significantly influenced by changes in vegetation type. Instead, a link between soil biogenic activity and the $\delta^{13}C$ of Ardèche stalagmites seems more likely. Thus, the significant decrease in $\delta^{13}C$ values at the beginning of the Holocene can be attributed to the resurgence of soil biogenic activity following post-glacial warming above the cave. Dissimilarities observed in the timing of the $\delta^{13}C$ variations in OR09-A and SM1-A $\delta^{13}C$ can be attributed to spatial differences in the Ardèche karst landscape, which is characterised by rugged terrain, steep slopes and abundant surface karstic dissolution features. These would give rise to intense spatial variations in soil thickness and development. Indeed, the $\delta^{13}C$ is likely primarily linked to local parameters such as soil activity and vegetation density above the cave, which would have undergone heterogeneous evolution across the plateau.

Magnesium and strontium concentrations in calcite generally reflect hydrological variations in the vadose environment above the cave or aerosol contributions (Dredge et al., 2013; Regattieri et al., 2014; Verheyden et al., 2000). The excessive residence time of percolation water in the karst, which for a given flow path is directly related to water balance, is known to influence speleothem trace element concentrations (Regattieri et al., 2014; Verheyden et al., 2000). However, long residence times leave their mark most prominently in dolomitic terrains due to incongruent dissolution processes (Fairchild et al., 2000). The lack of dolomite in Ardeche makes it unlikely that this process strongly influences the trace element variations in SM1-A.

Alternatively, PCP may explain the variations in magnesium concentrations. The incidence of PCP can significantly affect the percolation-water Mg/Ca during dry periods by reducing Ca concentrations along the flow path to the stalagmite, resulting in an increased calcite Mg/Ca (McMillan et al., 2005; Regattieri et al., 2014; Verheyden et al., 2000). The overall decrease in Mg/Ca observed in SM1-A during the Greenlandian and Northgrippian may thus be interpreted as a progressive increase in site moisture balance. This is supported by the pattern of $\delta^{13}C$ changes, particularly at the time of the large decrease in Mg/Ca in the early Holocene (~11.2–10.3 ka; Fig. 3), and suggests rapid soil and vegetation development, driven by optimal climatic conditions (warm temperature and sufficient moisture) at that time. In addition, the development of soil above the cave may have contributed to the reduction of the PCP effect by smoothing the infiltration rate. The progressive increase of moisture balance supported by our proxies through the first half of the Holocene is consistent with previous studies showing increased precipitation and humidity in northwestern Mediterranean region (e.g. Herzschuh et al., 2023; Morellón et al., 2018).

The possible contribution from exotic sources of Mg (and Sr), such as dust, must also be considered. A dust-sourced Mg contribution has been invoked to explain Ardeche speleothem Mg/Ca variations during the early Last Glacial period (Corrick, 2022). During the Greenlandian, as the wetter climate developed, residual dust-sourced Mg may have become progressively exhausted. However, if dust had a significant influence on Mg (and Sr) supply, we would expect a progressive change in the Mg/Sr ratio as the dust source was exhausted, from a mixed dust-plus-bedrock signature to a bedrock-only signature, due to the likely differences in the Mg/Ca and Sr/Ca ratios of the dust and local bedrock. Such a change in Mg/Sr is not observed, however.



When trace element concentrations are linked to hydrological changes via PCP, strong positive covariation between Sr and Mg is often present (e.g. Fairchild et al., 2000; McMillan et al., 2005; Regattieri et al., 2014), with a generally observed slope of ln(Mg/Ca) vs ln(Sr/Ca) ranging between 0.709 and 1.45 (Wassenburg et al., 2020). The Sr and Mg of SM1-A show a significant positive correlation (Table 3), and although the slope is below 0.7 (Fig. A3), the similar long-term trend between the two elemental ratios supports the hypothesis of an increase of moisture balance through the first part of the Holocene.

Growth rates can also influence Sr/Ca values by altering the partitioning coefficient (Gabitov et al., 2014). In the case of SM1-A, although there are some similarities between the growth-rate and Sr/Ca time series, the correlation coefficient is very low (Table 3). Similar to results from natural-cave and cave-analogue laboratory studies (e.g. Huang and Fairchild, 2001; Tremaine and Froelich, 2013; Wassenburg et al., 2020), the absence of a clear relationship between Sr/Ca and growth rate in SM1-A can be explained by a growth rate that is too low (i.e. less than 30 μm year$^{-1}$; Gabitov et al., 2014) and insufficiently variable to

significantly impact Sr partitioning. Drip-water Sr and Mg concentration from monitoring would be useful to further investigate the relationship between growth rate and Sr/Ca of Ardèche speleothem calcite (Huang and Fairchild, 2001).

        In summary, the evidence suggests that the variations in Mg/Ca and Sr/Ca of SM1-A are mainly linked to changes in site moisture balance. This is particularly evident at the beginning of the Holocene, when important climatic and environmental changes occurred (e.g. Baker et al., 2017; Bernal-Wormull et al., 2023; Dendievel et al., 2022; Wassenburg et al., 2016).

The δ$^{18}$O of speleothem calcite is primarily linked to the δ$^{18}$O of local precipitation and the cave air temperature during drip-water-to-calcite isotopic fractionation. Numerous mechanisms perturb the rainfall δ$^{18}$O signal between the moisture-source region and the cave (McDermott, 2004). At some sites, the calcite δ$^{18}$O can be interpreted more or less as a direct proxy for air temperature (e.g. Fohlmeister et al., 2012; Parker and Harrison, 2022). This is typically the case in the high latitudes or at high altitudes because fractionation during air-mass rainout and condensation is accentuated as temperature falls (Dansgaard, 1964).

However, in Ardèche, considering the local gradient between air temperature and precipitation δ$^{18}$O of +0.18 ‰ C$^{\circ-1}$ (Genty et al., 2014) and a drip-water-to-calcite fractionation of -0.18‰ C$^{\circ-1}$ in the cave (Tremaine et al., 2011), there is approximately zero net effect of local air temperature on the calcite δ$^{18}$O. The lack of resemblance between the δ$^{18}$O of our speleothems and some palaeotemperature reconstructions from southern France and Gulf of Lion (e.g. d'Oliveira et al., 2023; Jalali et al., 2016; Martin et al., 2020) supports the idea that regional air-temperature change was not the main forcing parameter for the δ$^{18}$O.

Considering the general trend of SM1-A and OR09-A δ$^{18}$O and the information provided by the other proxies, it is more likely that the δ$^{18}$O variations were related to hydrological changes.

        During the Greenlandian, warm temperature (strong Northern Hemisphere summer insolation) and increased humidity (decreasing Mg/Ca and Sr/Ca) allowed for the development of soil and vegetation above the caves, leading to decreasing speleothem δ$^{13}$C. Due to the summer water deficit and higher evapotranspiration rates characteristic of the Mediterranean

climate, vegetation development may have significantly reduced the proportion of summer rainfall contributing to karst recharge. Thus, the combined effect of rainfall seasonality and vegetation cover development could have reduced the proportion of isotopically heavier summer rainfall contributing to the annually integrated δ$^{18}$O of the karst recharge, potentially explaining the observed decrease in calcite δ$^{18}$O during the Greenlandian. Finally, the Mediterranean influence is also noted by the fact that the absolute values of δ$^{18}$O in our stalagmites are similar to other Mediterranean speleothems (Bernal-Wormull

et al., 2023; Genty et al., 2014; McDermott et al., 1999), for which the input of Mediterranean-sourced precipitation has been invoked to explain higher values than those of other European archives situated beyond Mediterranean influence (Affolter et al., 2019; Fohlmeister et al., 2012; Waltgenbach et al., 2020).





### 4.2 The 8.2 ka event in southern France: a subtle event or an archival issue?

The high-resolution stable isotope and trace element time series of SM1-A and OR09-A do not reveal any significant change in Ardèche during the 8.2 ka event time frame. The lack of an unambiguous, prominent variation points to a limited climatic impact of the 8.2 ka event in southern France. This would be consistent with the absence of clear responses recorded by other southern French stalagmites (Genty et al., 2006; McDermott et al., 1999), pollen records (Jouffroy-Bapicot et al., 2007; Martin et al., 2020; d'Oliveira et al., 2023), or marine sediment cores (Jalali et al., 2016), but it seems to be at odds with palaeoenvironmental archives recording changes during the 8.2 event in western Europe. Some of these records were interpreted in terms of temperature decrease (e.g. Boch et al., 2009; Von Grafenstein et al., 1999; Tinner and Lotter, 2001). The low sensitivity of the investigated proxies of SM1-A and OR09-A to temperature changes, particularly as they were likely subtle (~-0.5°C during the 8.2 ka event in the Mediterranean basin; Morrill et al., 2013; Wiersma and Renssen, 2006), may explain the absence of clear 8.2 ka event signal in Ardeche stalagmites.

Other studies (e.g. Bernal-Wormull et al., 2023; Kilhavn et al., 2022; Waltgenbach et al., 2020) have interpreted the $\delta^{18}O$ decrease associated with the 8.2 ka event as a change in the isotopic composition of the vapour source (i.e. freshening of the Atlantic Ocean surface; Ellison et al., 2006), which is supported by models simulating the $\delta^{18}O$ signature of the event in precipitation (Holmes et al., 2016; Tindall and Valdes, 2011). Such a $\delta^{18}O$ change is not recorded in SM1-A and OR09-A, although Ardèche is relatively close to the Atlantic Ocean and falls under the influence of the mid-latitude westerlies. This may be explained by the significant proportion of Mediterranean-sourced precipitation in the region, characterised by a higher $\delta^{18}O$ signature and thus cancelling any signal carried by Atlantic moisture at the time of the 8.2 ka event (Celle-Jeanton et al., 2001; Genty et al., 2014).

The 8.2 ka event in Europe has also been associated with a drier climate (e.g. Allan et al., 2018; Baldini et al., 2002), which is supported by model-data comparison (Wiersma and Renssen, 2006) and interpreted as the result of a decrease in the strength of the westerlies and their shift southward (e.g. Benson et al., 2021; Holmes et al., 2016). The lack of a significant change at 8.2 ka recorded by our speleothem proxies could thus also be explained by Ardèche being situated in a less sensitive position to a westerlies shift. Indeed, if a southward shift occurred during the 8.2 ka event, it is likely that southern France was still under the westerlies path, but at its northern limit (Wassenburg et al., 2016). Thus, being in the transition zone, Ardèche climate may not have experienced a drastic change to a drier climate. This supports the finding of Tindall and Valdes (2011), who showed that the 8.2 ka event effect on precipitation change seems to have been geographically variable, and needs to be considered as a local rather than a regional marker (Tindall and Valdes, 2011).

At a local scale, the Ardèche climate may have experienced an increase in the frequency of intense hydrological events, as supported by phytolytic observations at two sites near the caves (i.e. Lalo, ~40 km and Les Brassières, ~25 km away; (Berger et al., 2016; Delhon, 2005). These events notably appear during a period of frequent fires and increased Mediterranean-sourced precipitation (Berger et al., 2016). This is consistent with the small peak between 8.2 and 8.1 ka observed in the $\delta^{18}O$ of both SM1-A and OR09-A, which can reflect intensified summer rainfall events sourced from the Mediterranean Sea (e.g. as the aforementioned "Cevenol" rainfall episodes). Although the Lalo and Les Brassières studies highlight that this was not sufficient to overturn the local vegetation, these events may have significantly increased soil leaching and erosion (Berger et al., 2016; Delhon, 2005), which could have triggered the slight $\delta^{13}C$ peak in OR09-A. However, since these $\delta^{18}O$ and $\delta^{13}C$ oscillations are not significant when viewed against the millennial periods either side, a confident conclusion about these variations is difficult to reach.

Thus, the result of this study testifies that despite their potential for high-resolution analysis, individual stalagmite proxy signals always reflect the integration of several—often local—forcing parameters, some of which may have countering effects. This



highlights the need for complementary approaches through the cross comparison of multiple archives in order to decipher regional palaeoclimatic responses to the 8.2 ka event.

**5. Conclusion**

Our multiproxy analysis ($\delta^{18}O$, $\delta^{13}C$, Mg/Ca, Sr/Ca) of stalagmites from St Marcel Cave and Aven d'Orgnac in southeastern France does not reveal a significant climatic anomaly that can be linked to the 8.2 ka event, in spite of the caves being situated

in close proximity to the Atlantic Ocean and falling under the influence of the westerlies. A lack of sensitivity to air temperature changes by the stalagmites in this region may explain the absence of a clear response of the calcite $\delta^{18}O$ to the event. A possible Mediterranean influence may have also contributed to attenuating or masking the effects of any North Atlantic freshening from this event on the $\delta^{18}O$ signal, but it does not explain the absence of a hydrological response from the Mg/Ca and Sr/Ca. Whilst it is well known that these ratios do not always yield meaningful hydrological information at all cave sites, the positive

covariance of Mg/Ca and $\delta^{13}C$ over the whole 11-5.5 ka interval suggests SM1-A was sensitive to hydrological change during its growth history, at least over the long term. This finding underscores the importance of multiproxy analyses to deconvolve climatic information and reduce interpretation errors.

The results of this study contrast with European palaeoenvironmental records further afield from the northwestern Mediterranean basin, where the 8.2 ka event is clearly marked. Nevertheless, the SM1-A and OR09-A records align with

several other studies from southern France that also report no significant changes. This questions the generalised nature and impacts of the 8.2 ka climatic event. Modelling coupled with data provided by a range of archive types could help reconcile regional discrepancies between proxy records, provided that biases introduced by different archives/proxies and site effects are considered Providing new high-resolution and precisely dated data from regions with complex hydrological regimes, like southeastern France, would be a considerable asset for better understanding the impacts of this event.






**Appendices**

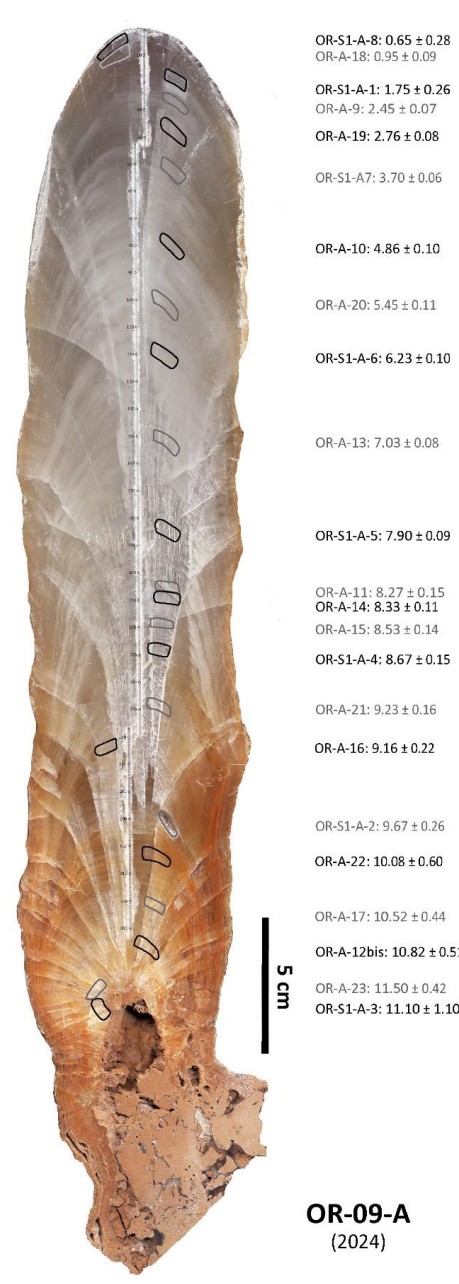

**Figure A1.** Polished section of OR09-A stalagmite (Aven d'Orgnac, France). At right, the U-Th dating samples and their ages (± 95% uncertainties) are indicated (ka BP; colour switching better facilitates matching age and sample position).




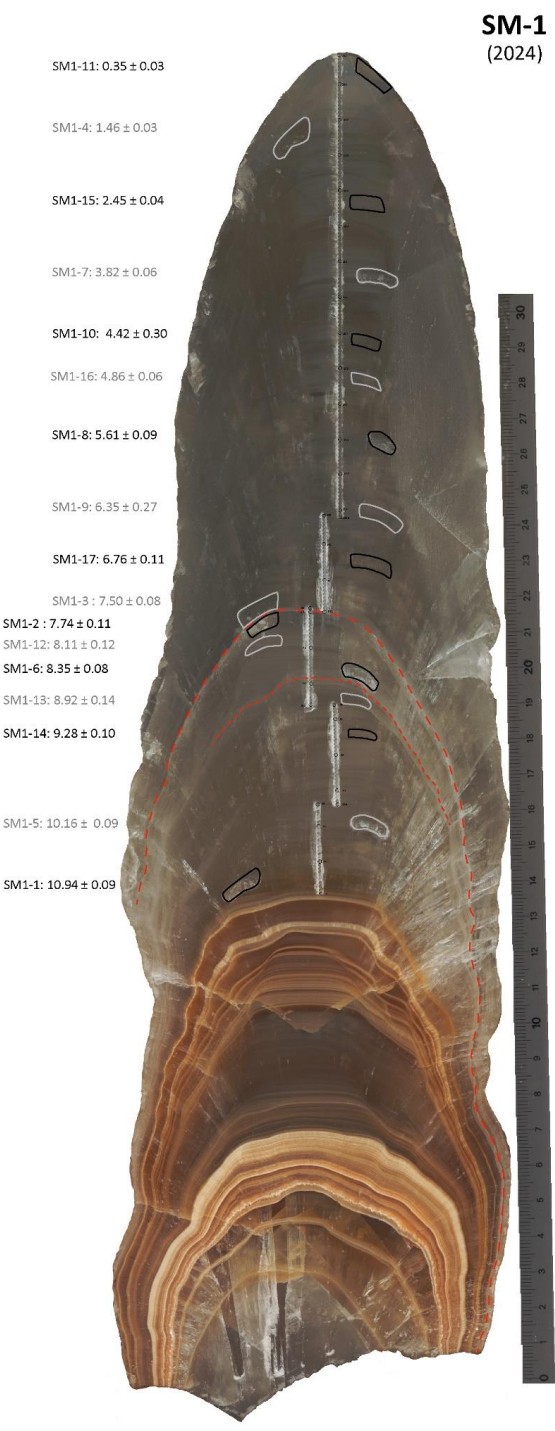

**Figure A2.** Polished section of SM-1 stalagmite (St-Marcel Cave, France). At left, the U-Th dating samples and their ages (± 95% uncertainties) are indicated (ka BP; colour switching better facilitates matching age and sample position).



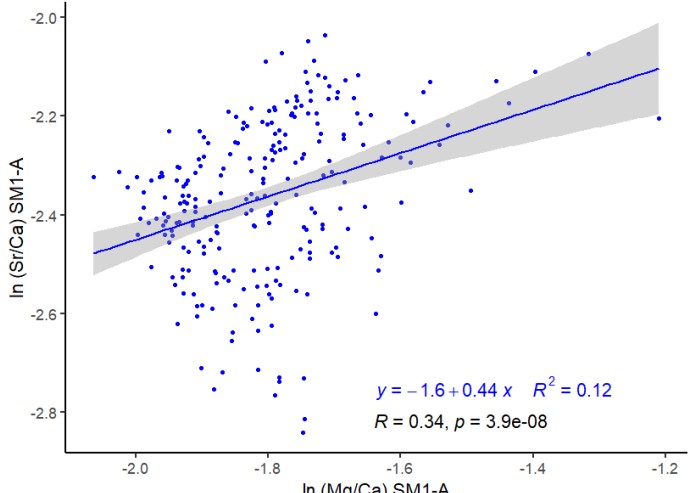


**Figure A3.** Simple linear regression between SM1-A ln (Mg/Ca) and ln (Sr/Ca). The equation of the regression line (slope = 0.44 ± 0.15) and the coefficient of determination ($R^2$) are indicated in blue, while the Pearson correlation coefficient (R) and p-value ($p$) associated with the linear model are indicated in black. The shaded grey area corresponds to the 95% confidence interval of the linear model.

### Author contribution

IC conceived the project, and prepared and sampled the speleothems for analyses. RD carried out the stable isotope ratio measurements. JH and DH made the U-Th measurements. AG carried out the trace element analyses. MP analysed and interpreted the data with guidance from IC and RD. MP prepared the manuscript, with input from IC and RD. All authors edited the final version.

### Competing of interests

Some authors are members of the editorial board of journal CP.

### Acknowledgements

We sincerely thank the entire teams at Aven d'Orgnac and St Marcel Cave for their warm welcome during our various field visits and for their interest in this work, especially Stéphane Tocino, Françoise Prud'homme, and Delphine Dupuy. We are also grateful to colleagues and students who occasionally lent a helping hand in the field or provided some of their time by 410 participating in sample preparation in the various laboratories: Didier Cailhol, Ellen Corrick, Stéphane Jaillet, Petra Bajo, Serene Paul, Timothy Pollard, Jérémie Gaillard, Maeva Monnier and Cloé Inzaina.

### Financial support

EDYTEM lab supported the master internship of MP.



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
