# Peer review of "The elusive 8.2 ka event in speleothems from southern France"

_EGUsphere, 2025_

## Author Comment (AC3)

We sincerely thank the three anonymous reviewers for their valuable feedback and suggestions. The reviewers' recommendations have been carefully addressed, and our responses are shown in green.

**Response to Anonymous Referee#1**

This manuscript presents first data from two stalagmites in southern France covering the Holocene period with the main aim of understanding why the 8.2 ka event is not present in paleoclimate records from the Mediterranean, particularly in those from southern France. The authors obtain the data to support this hypothesis from two stalagmites with robust chronologies and high-resolution isotopic profiles (trace elements are analyzed in one of the two stalagmites) from two different caves: Orgnac and St Marcel. The objective is sound and the methodology is adequate but the study lacks support from other Mediterranean records to the central hypothesis. In my opinion, major changes are required before publication (see below) and obtaining some monitoring data may be of interest too to better understand the response of these two caves to current climate.

INTRODUCTION AND MAIN HYPOTHESIS. the introduction should be completed with more references, particularly from marine cores from W Mediterranean. Evidences of the 8.2 ka event have been found in marine sediments from W Mediterranean, both in the Balearic Sea (Frigola et al., 2007) and in the Alboran Sea (Cacho et al., 2002), indicating a clear signal of a cold event and with intensified dry westerly winds. In fact, it has been proposed that this event interrupted the sapropel 1 (ORL in W Med) since the Western Mediterranean deepwater convection was reactivated. This idea should be incorporated in the introduction and later in the discussion because it contradicts the main hypothesis of the manuscript about the lack of signal of the 8.2 ka event in the Mediterranean realm.

We have reworked the introduction to include W Mediterranean ocean records. Unfortunately, we do not fully agree with the statement of Anonymous Referee #1. In the references cited, Cacho et al. (2002) suggest that the last ORL in the Alboran Sea (core MD95-2043) ends at the onset of the 8.2 ka event, and propose that stronger westerly winds were associated with the 8.2 ka event. However, the ORL presented in this paper abruptly ends around 9 ka, which is 800 years older than the onset of the 8.2 ka event. Therefore, the link between the two events cannot be substantiated as the ocean-core chronology is poorly constrained though the interval of interest (no age controls between 8.98 and 7.40 ka). Cacho et al. (2002) also present a SST record (alkenone-derived) of MD95-2043, showing a temperature decrease with minimum values centred at 8.2 ka (original chronology), corresponding to ~8.0 ka on a chronology based on the Marine20 calibration (our calculations). Even though the authors link this SST decrease to the 8.2 ka event, it must be noted that the excursion starts at ~9 ka (chronologically constrained by a 14C date), which is significantly older than the onset of the 8.2 ka event (8. 25 ka, Thomas et al., 2007) and implies an event duration of ~800 years in contrast to the 160 years of the 8.2 ka event (Thomas et al., 2007). In addition, in a more recent study, SSTs from the same ocean sediment core have been re-evaluated (G. bulloides Mg/Ca-derived), and do not show any significant temperature shifts around 8.2 ka (Català et al., 2019). In the absence of consistency between the two SST records, and without precise chronological constraints, it is difficult to confidently determine whether a cooling related to the 8.2 event is recorded in MD95-2043.

The second reference cited, Frigola et al. (2007), presents an event of intensified westerly winds occurring during the 9.0–7.8 ka period (called the M8 event) and recorded in the Balearic Sea core MD99-2343. Even though the M8 event encompasses the 8.2 ka event interval, it cannot be reconciled with an atmospheric response to a short-term (~160 yr) freshwater discharge entering the North Atlantic at 8.2 ka BP. In support of this, a more recent study presents new SST data from the same core (MD99-2343), which do not show significant surface temperature changes during the 8.2 ka event period (Català, 2019).

Finally, the Western Mediterranean core ODP-976 (Alboran Sea, Martrat et al., 2014) presents a short cold SST anomaly; however, its timing prevents it being tied unequivocally to the 8.2 ka event. The event occurred around 7.5 ka (original chronology; or 7.2 ka based on the Marine20 calibration), which is considerably younger than the timing of the 8.2 ka event. The authors highlight the absence of age control on this event, which complicates the characterisation of the timing and duration of the event recorded in ODP-976, and thus its potential link with the 8.2 ka event.

We would also like to highlight the absence of significant variations in the proxies of North Atlantic core MD01-2444 (Portuguese margin, near the Strait of Gibraltar, Hodell et al., 2013), which calls into question any likelihood of Western Mediterranean cores recording the 8.2 ka event.

In summary, from what we can determine, local ocean records either lack the signal or lack a suitably precise chronology to support the detection of the 8.2. ka event.

It is also very important to show the records compared to the two stalagmites; Fig. 1 with the map including so many records is really nice and informative but the records themselves should be also incorporated as a final figure, clearly indicating which records are under the influence of Atlantic climates, which are under Mediterranean and which show an annual signal versus those biased towards one season. In that different way of incorporating the climate signal may be the key to understand the 8.2 ka event in a regional scale. The manuscript, as it is now, lacks the comprehensive view required to explore the proposed hypothesis.

A figure presenting the different records cited in the manuscript will be added. This figure will show the records by geographical region, which helps with the identification of the influence of Mediterranean- and Atlantic-sourced rainfall, and the characterisation of the variable 8.2 ka event impacts over Europe.

MONITORING. I understand that the drip-water was monitored in a cave that was close to Orgnac cave (line 105) but that does not guarantee the same pattern in drip-water or the same infiltration processes in the two studied caves, not over the two different stalagmites. The amount of soil, the thickness of the host rock or even the type of rock and its fracturation pattern are other important factors that may condition drip water dynamics in Orgnac cave that may not be the same in St Marcel cave.

In fact, the authors use those factors related to soil processes to justify the different pattern in d13C in both stalagmites (line 200). Both caves may be under different environmental conditions, i.e. soil activity, epikarst PCP, etc, that lead to such diverse isotopic profiles (d13C). Similarly, those different conditions may require conducting monitoring surveys in the cavities to fully understand dripwater processes, relation with rainfall patterns, etc.

We agree that monitoring can provide useful additional information for identifying the processes affecting proxies recorded in speleothems. However, to be reliable, a cave must be monitored over a substantial number of years, sufficient to incorporate a reasonable 'sample'

of interannual climate variability. In the cases of St. Marcel and Orgnac Caves, no monitoring was implemented because the speleothems collected in St. Marcel Cave were inactive, broken and not necessarily in situ, and whilst OR09-A was in situ, it was inactive at the time of collection. Under these circumstances, a monitoring of the drip water feeding these stalagmites cannot be carried out.

Moreover, as highlighted in our paper and in the Anonymous Referee#1's comments, drip water dynamics and composition are influenced by numerous local factors and vary not only from cave to cave, but also between chambers or even fissures within the same cave (Fairchild and Baker, 2012). Unless the speleothems being studied are situated beneath the drip points being monitored, this variability can lead to monitoring data being unreliable when interpreting specific speleothem proxy records, and therefore it cannot be considered as essential for the interpretation of a speleothem paleoclimatic signal. Many, many speleothem studies have been published without supporting information from cave monitoring.

INTERPRETATION OF PROXIES. Comparison of the two d18O or the two d13C profiles does not help too much to interpret the meaning of those proxies in these particular caves and time period. I wonder if providing trace elements for the two stalagmites may be of further help.

This comment has inspired us to search for the extra resources needed to implement these analyses. The OR09-A trace element measurements have now been completed and new data will be added to the results and discussion of the manuscript.

Now, both trace elements (Mg and Sr) are quite different and they do not compare well with d13C to be explained all proxies in terms of hydrological variations. In general, my feeling is that the two stalagmites are well-dated but they do not respond in a similar way to past climate/environment conditions. The reasons remain elusive but the authors may want to explore more and complete the section about proxy interpretation (section 4.1) with more ideas or more data. As it is now, it is not very convincing.

We believe that the interpretation of the proxies was addressed in a comprehensive and nuanced manner, with all the likely candidate processes accounted for. The different proxies are indeed sensitive to hydrology (and are interpreted accordingly), but they respond to different processes. For example, one of the main parameters influencing  $\delta^{18}O$  of the studied speleothem is the distribution of Mediterranean versus Atlantic rainfall. In contrast, the  $\delta^{13}C$  is more influenced by processes occurring in the soil above the cave, which are themselves linked to precipitation and temperature, as well as within the cave itself (e.g. through PCP). Finally, trace elements are sensitive to processes in the bedrock, dominated by PCP (prior calcite precipitation). Although all of these processes are related to hydrology to varying degrees, they help explain the differences recorded by the proxies in this study.

**Response to Anonymous Referee#2**

The 8.2 ka event is widely regarded as the most significant climate shift in the N Hemisphere since the end of the Younger Dryas. And it is probably the best studied event in the Holocene whose timing and origin as well understood. Still, there are regions, in particular close to the Mediterranean Sea, where this event has not been recorded, even in high-resolution archives. The present ms. adds new observational evidence from two cave sites in S France showing no significant proxy response to this N Atlantic cooling and drying event.

The ms. is well written and the proxy evidence is fairly convincing. I regret, however, that despite the authors' claim about the importance of multiproxy analyses (line 376) they do not present trace element data for the second stalagmite. If possible, I recommend to add Mg+Sr data at least across the critical time range of the 8.2 ka event.

See previous response to Referee#1 – trace element data for OR09-A will be presented in the revised ms.

Another aspect that could/should be improved is the nearly complete lack of petrographic information. The reader is shown the scans of the slabs and somewhere I read a sentence about the fabric, but nothing else.

Petrographic information was not developed thoroughly in this study as we felt there was nothing relevant to add to the core of this paper. Obviously, we carefully examined the polished sections of both speleothems and observed that they exhibit a fairly homogenous fabric all the way along their growth axis, with only minor and subtle changes in colour or microporosity of the calcite (i.e. laminae being more or less visible). Therefore, we consider it overly destructive to make thin sections on a speleothem where visible, substantive fabric changes are lacking. However, we will add more descriptive information in the manuscript about fabrics.

**Minor:**

Line 49: Herbstlabyrinth Cave

• Line 53: Schleinsee

• Fig. 2A: explain dashed lines

Thank you, this has been corrected.

**Response to Anonymous Referee#3**

Passelergue et al. present proxy data for two speleothems from southern France, which do not show any obvious peaks during the 8.2 ka event, which is considered as one of the major climate anomalies of the Holocene. They interpret this pattern as evidence that the 8.2 event did not have a major impact in southern France or at least as the absence of a clear hydrological response.

The paper is interesting and well written, and I agree with the general conclusion and can recommend publication in CP. However, I do not fully agree with the interpretation of the proxies (e.g., the Mg and Sr data in terms of PCP, see detailed comments below). In addition, I think that the authors may have a bit more to offer. Based on the references, it is clear that they compiled all the literature for the 8.2 ka event in Europe. (i) It would be good to see at least some of these records in a figure to highlight the (regionally consistent?) similarities and differences, which are currently only described in the text. (ii) This comparison may allow to draw some more general conclusions about the expression of the 8.2 ka event in proxy records from Europe (i.e., in which archives/proxies does it occur?) and its (general or spatially variable?) climatic features (cold, dry, wet, etc.).

A figure presenting some of the available records cited in the manuscript has been added. Interpretation in terms of climatic impact of the 8.2 ka event has been incorporated as much as possible, while remaining within the bounds of what the authors provided.

**Detailed comments:**

Fig. 1: See my general comment. The figure shows various sites/proxy records that obviously were compiled for the paper. It would be really useful to see a comparison with at least a few of those in a figure. Furthermore, it may be possible to derive some more general information about the expression of the 8.2 ka event in Europe, beyond southern France.

**See response above.**

Line 63: 'However, some records suggest that the presence of sapropel S1, which occurred at the same time, masks the possible influence of the 8.2 ka event in some regions (e.g. Magny et al., 2007; Siani et al., 2013; Zanchetta et al., 2007). More generally, there are no high-resolution records in the western Mediterranean region showing significant changes that can be clearly attributed to the 8.2 ka event (e.g. d'Oliveira et al., 2023; Jalali et al., 2016; McDermott et al., 1999; Fig. 1a, b).' Should the order of these two sentences be changed? Otherwise, I do not get the meaning of the first one.

**This paragraph has been modified.**

Line 107: 'Since the host rock and fracture network are similar at all sites, we can expect the same mixing of infiltration waters at St Marcel and Orgnac Caves. This implies that the  $\delta^{18}O$  of the calcite represents at least an intra-annual, if not an inter-annual, average of the  $\delta^{18}O$  of recharge precipitation.' Although I agree with the reasonable conclusion in the second sentence, I think, the interpretation in the first sentence is too 'optimistic'. It is well known that even two nearby drip sites in the same cave can have very different characteristics.

We thank Referee #3 for pointing out this issue. This sentence was confusing, and will be corrected. Since the conditions that can potentially lead to a difference between annual rainfall  $\delta^{18}$ O and speleothem calcite  $\delta^{18}$ O (e.g., air temperature, vegetation, soil type and thickness) are similar between Chauvet Cave and Orgnac/St. Marcel Caves, one can expect the same composition of recharge water for all three caves. Genty et al. (2014) found no seasonal bias that could cause a difference between rainfall  $\delta^{18}$ O (Orgnac) and calcite  $\delta^{18}$ O (once water-calcite fractionation and cave temperature is taken into account). However, it must be noted that karst recharge is biased towards autumn and winter rainfall (i.e. rainfall is significantly higher during these seasons; Genty et al., 2014), which implies that speleothem proxies may be more representative of these seasons.

Whilst identical degrees of interannual mixing of drip waters at Orgnac/St. Marcel caves cannot be inferred from Chauvet drip monitoring (Genty et al., 2014), the time resolution of our trace-element and stable-isotope data implies that each proxy value integrates several years (~5 to 80 years)

Table 1: Please provide the uncertainties for the (230Th/232Th) activity ratios. In addition, it would be good to report a consistent number of significant digits.

The (230Th/232Th) activity ratio is not used in age calculations because mathematically it is the (232Th/238U) activity ratio that is required to calculate a corrected age (e.g. equation 2 of Cheng et al., 2000). We provide the redundant 230Th/232Th ratio (derived from (230Th/238U) and (232Th/238U) ratios) only as a convenience for the reader, given its long history in qualitative assessment of the likely impact of detrital Th contamination. For any specialist reader with a requirement for (230Th/232Th) uncertainty, it can be determined from the uncertainties of (230Th/238U) and (232Th/238U) ratios provided.

We thank Referee #3 for pointing out the inconsistency in the digits (significant figures) reported in Tables 1 and 2. We have rectified this for the final ages of SM1-A and OR09-A to

two digits. However, enforcing a constant number of decimals for  $^{238}$ U in OR09-A would either reduce the measurement precision (e.g.  $50.0 \pm 0.2$  to  $50 \pm 0$ ) or create artificial precision (e.g.  $42 \pm 8$  to  $42.0 \pm 8.0$ ).

Line 172: 'The empirically derived (230Th/232Thinitial) corrections applied were 0.48±0.32 for SM1-A and 0.48±0.24 for OR09-A.' Why are the uncertainties different? I guess this is related to the method used to derive the corrections, but a brief explanation would be good.

Since SM1-A is very clean, we encountered difficulties in the determination of the best initial Th correction to apply. For these reasons, and because Orgnac cave is located nearby in the same watershed and within the same bedrock, we applied the initial Th correction of OR09-A to SM1-A, but with larger uncertainties. The difference on the calculated ages is negligible. For illustration, a comparison table of the SM1 dates using the two different corrections is provided below. We thank Referree#3 for raising this point; we will add an explanation of this difference in the manuscript.

Table: SM1-A U-Th dates (age in ka 1950 BP) calculated using the initial Th corrections of  $0.48 \pm 0.32$  and  $0.48 \pm 0.24$ .

| Sample ID | Age ka BP (Initial Th correction of $0.48 \pm 0.32$ ) | Age ka BP
(Initial Th correction of 0.48
± 0.24) |
|-----------|-------------------------------------------------------|--------------------------------------------------------|
| SM1-11    | $0.35 \pm 0.03$                                       | $0.35\pm0.03$                                          |
| SM1-4     | $1.46\pm0.03$                                         | $1.46\pm0.03$                                          |
| SM1-15    | $2.45\pm0.04$                                         | $2.46\pm0.04$                                          |
| SM1-7     | $3.82\pm0.06$                                         | $3.82\pm0.06$                                          |
| SM1-10    | $4.42\pm0.30$                                         | $4.43\pm0.30$                                          |
| SM1-16    | $4.86\pm0.06$                                         | $4.86 \pm 0.06$                                        |
| SM1-8     | $5.61\pm0.09$                                         | $5.62\pm0.09$                                          |
| SM1-9     | $6.35\pm0.27$                                         | $6.35\pm0.26$                                          |
| SM1-17    | $6.76\pm0.11$                                         | $6.76\pm0.11$                                          |
| SMI-3     | $7.50\pm0.08$                                         | $7.50\pm0.08$                                          |
| SM1-2     | $7.74 \pm 0.11$                                       | $7.74 \pm 0.11$                                        |
| SM1-12    | $8.11\pm0.12$                                         | $8.11\pm0.11$                                          |
| SM1-6     | $8.35\pm0.09$                                         | $8.35 \pm 0.09$                                        |
| SM1-13    | $8.92\pm0.14$                                         | $8.92\pm0.15$                                          |
| SM1-14    | $9.28\pm0.10$                                         | $9.28 \pm 0.10$                                        |
| SM1-5     | $10.16\pm0.09$                                        | $10.16\pm0.08$                                         |
| SM1-1     | $10.94 \pm 0.08$                                      | $10.95\pm0.09$                                         |

Section 3.2: 'Isotopic Equilibrium Conditions' I think the header is misleading because other aspects influencing the isotope data are discussed here as well. In addition, this section should in my opinion be moved into the discussion section or at least not be discussed prior to the general description of the data (section 3.3). I would suggest to integrate the whole section into section 4.1.

**We thank Referee #3 for this suggestion. We will apply it to the corrected manuscript.**

Line 181: 'However, the absence of consistent covariation between  $\delta^{13}C$  and  $\delta^{18}O$  in each of the stalagmites (Fig. 3) indicates that calcite precipitation was not associated with significant kinetic fractionation.' The d18O and d13C values of stalagmite SM1-A are significantly correlated (Table 3), so this statement is not correct. However, there are numerous speleothem studies showing highly correlated d18O and d13C values due to climatic reasons rather than disequilibrium isotope fractionation that the absence of a correlation should – in my opinion – not be interpreted as evidence for the lack of in-cave effects. PCP, for instance, would also result in a correlation between d18O and d13C values. In general, I do not understand why this topic is discussed in such detail. The major point of the paper is that there is not strong signal during the 8.2 ka event. This interpretation does not require equilibrium isotope fractionation. Quite the contrary, very dry conditions, for instance, could result in low drip rates supporting disequilibrium fractionation and positive peaks in d13C and d18O, which would even be useful.

The correlation coefficient between  $\delta^{18}O$  and  $\delta^{13}C$  in SM1-A is indeed statistically significant, with a value of 0.44, but this has low explanatory power, and cannot be considered as fully supporting the presence of consistent covariation between them. We acknowledge and agree with the second part of Referee #3's comment. While the presence of covariation between  $\delta^{18}O$  and  $\delta^{13}C$  may not necessarily imply disequilibrium isotope fractionation (but may reflect climatic parameters affecting both proxies), in the present case, the absence of strong covariation between the two proxies supports the hypothesis that disequilibrium fractionation was not the dominant process.

Line 193: 'For most of the growth interval, the  $\delta^{18}O$  signals of these stalagmites show similar trends with a limited amplitude (Fig. 4a). ... Indeed, besides some discrepancy during the periods 9.7–9.3 ka and 8.0–7.6 ka, the two  $\delta^{18}O$  patterns are very similar.' I agree. To further support this statement, it may be useful to calculate (running) correlations between the two curves.

Thank you for suggesting this; it has been calculated and added to the manuscript.

Line 195: 'This is in spite of an offset for much of the period, with OR09-A having generally higher values (Fig. 4a and b).' Looking at Fig. 4, which is very useful, it seems to me that there is not just an offset, but also some centennial scale differences of more than 0.5 permil. This may be related to dating uncertainties, but could also be due to local effects occurring at the surface or in the karst aquifers and caves.

Indeed, these small differences can be due either to dating uncertainties or to local effects. We can't determine which of these reasons is the right one - maybe it is a bit of both. As it is not core to the paper, we chose to not discuss it.

Table 3: The high correlation coefficient between Mg/Ca and Sr/Ca is really surprising to me, not only from Fig. 3, but in particular considering Fig. A.3, which just shows a point cloud.

This is likely because Figure A.3 was based on both early/middle Holocene data presented in our paper, and late Holocene data, which are not presented in this paper. We have replaced this figure with one using data from the period 11.5–5.5 ka only, to ensure greater consistency.

We would also like to emphasise that the correlation coefficient for this time interval between SM1-A Mg/Ca and Sr/Ca ( $\rho$  = 0.46) should not be considered high, but rather weak to moderate.

Line 247: The introductory paragraph to the discussion section can be deleted.

**This paragraph will be deleted.**

Line 262: 'Magnesium and strontium concentrations in calcite generally reflect hydrological variations in the vadose environment above the cave or aerosol contributions (Dredge et al., 2013; Regattieri et al., 2014; Verheyden et al., 2000).' I strongly disagree with this statement. There are many speleothem trace element records, where both Mg and Sr do not show a robust relationship with any climate parameter. I agree that – if e.g. PCP can be demonstrated to have an effect – both elements are powerful hydrological proxies. However, this is not 'generally' the case, and then, the interpretation in terms of palaeoclimate is often very difficult.

**We agree that this sentence should be modified to include more nuance.**

Line 268: 'Alternatively, PCP may explain the variations in magnesium concentrations.' Yes, if the effect can be demonstrated, for instance with a positive Sinclair test (see below). If this test is negative, PCP does probably not have a major influence and other processes need to be considered. Since the Sinclair test is negative, I think that the interpretation in terms of PCP (and hydrology) goes to far. However, as stated above, this is not a problem for the major conclusion of the paper about the 8.2 ka event.

We agree with Referee #3 statement, and this review has convinced us to interrogate more deeply SM1 trace element variations, and revise our interpretation accordingly. A Sinclair test applied to different time slices allows us to reveal a slope of 0.88 for the 11-6.5 ka period, which supports (see figure below) PCP as the main driver of Mg/Ca and Sr/Ca variations for a wide interval encapsulating the 8.2 ka event.

Before 11 ka and after 6.5 ka, other processes likely had a significant influence on the trace element ratio in SM1-A and altered the Sinclair's test slope. Since Mg/Ca and Sr/Ca variations before 11 ka and after 6.5 ka are not the primary concern of this paper (i.e. the 8.2 ka event), a more detailed study of the factors influencing the trace elements of SM1-A will be addressed in a paper currently in preparation.

Figure: Sinclair's test performed on SM1-A Sr/Ca and Mg/Ca for the 11.5-5.5 ka period (slope and equation in blue) and the 11-6.5 period (slope and equation in red). Data removed for this

second test are indicated by triangles. Colours represent the corresponding age of the trace element values.

Line 287: 'The Sr and Mg of SM1-A show a significant positive correlation (Table 3), and although the slope is below 0.7 (Fig. A3), the similar long-term trend between the two elemental ratios supports the hypothesis of an increase of moisture balance through the first part of the Holocene.' It would be good to demonstrate this 'similar long-term trend', e.g., by calculating the correlation between two strongly smoothed curves or linear fits through both proxies.

We thank Referee #3 for this suggestion and will follow this recommendation.

Line 297: 'In summary, the evidence suggests that the variations in Mg/Ca and Sr/Ca of SM1-A are mainly linked to changes in site moisture balance.' I do not agree with this statement (see above). However, this interpretation of the Mg and Sr data is not crucial for the major conclusion of the paper.

A deeper analysis has been performed to better understand trace elements variations. See the response above.

Line 310: 'Considering the general trend of SM1-A and OR09-A  $\delta^{18}O$  and the information provided by the other proxies, it is more likely that the  $\delta^{18}O$  variations were related to hydrological changes.' Or they are not at all sensitive to climate change, at least during the Holocene when climate conditions at the surface were rather stable. Even the largest event of the Holocene, the 8.2 ka event (if so), was not recorded.

Through the presented interval, the amplitude of  $\delta^{18}$ O is >1‰, which is significant enough to invoke changes in the environment of the cave or in the characteristics of the water feeding the stalagmite. Hence, we did not consider it appropriate to state that the  $\delta^{18}$ O variations may not be at all sensitive to some aspect of climate change.

Section 4.2: See my general comment. I agree with everything in this section, but I believe that the authors may have more to offer. As the state in their last sentence, there is a '... need for complementary approaches through the cross comparison of multiple archives in order to decipher regional palaeoclimatic responses to the 8.2 ka event.' It seems to me that they already compiled this information and have everything available to perform such a cross comparison. The paper would strongly benefit from making more use of these data.

We thank Referee #3 for this suggestion. We will integrate a regional comparison of the impact of the 8.2 ka event in the discussion.

Line 373: '... but it does not explain the absence of a hydrological response from the Mg/Ca and Sr/Ca.' See my comment above. If Mg/Ca and Sr/Ca are not affected by PCP (as suggested by the negative Sinclair test), they do not reflect hydrological changes above the cave. Maybe, they are not sensitive to climate change (of this scale) at all. This statement is too ambitious to me.

See response and figure above; a positive Sinclair test has been identified for the period 11 - 6.5 ka, suggesting that the PCP is the main driver of Mg and Sr variations in SM1-A for this period.

Line 373: 'Whilst it is well known that these ratios do not always yield meaningful hydrological information at all cave sites, the positive covariance of Mg/Ca and  $\delta^{13}$ C over the whole 11-5.5 ka interval suggests SM1-A was sensitive to hydrological change during its growth history, at least over the long term.' I absolutely agree with the first part of the sentence (see my previous

comments). With the second part, I do not agree considering the negative Sinclair test, Fig. A.3 and Fig. 3.

See responses above.

**References:**

Cacho, I., Grimalt, J. O., and Canals, M.: Response of the Western Mediterranean Sea to rapid climatic variability during the last 50,000 years: a molecular biomarker approach, Journal of Marine Systems, 33–34, 253–272, https://doi.org/10.1016/S0924-7963(02)00061-1, 2002.

Català, A., Cacho, I., Frigola, J., Pena, L. D., and Lirer, F.: Holocene hydrography evolution in the Alboran Sea: a multi-record and multi-proxy comparison, Climate of the Past, 15, 927–942, https://doi.org/10.5194/cp-15-927-2019, 2019.

Fairchild, I. and Baker, A.: Speleothem Science: From Process to Past Environments, i pp., https://doi.org/10.1002/9781444361094.app1, 2012.

Frigola, J., Moreno, A., Cacho, I., Canals, M., Sierro, F. J., Flores, J. A., Grimalt, J. O., Hodell, D. A., and Curtis, J. H.: Holocene climate variability in the western Mediterranean region from a deepwater sediment record, Paleoceanography, 22, https://doi.org/10.1029/2006PA001307, 2007.

Genty, D., Labuhn, I., Hoffmann, G., Danis, P. A., Mestre, O., Bourges, F., Wainer, K., Massault, M., Van Exter, S., Régnier, E., Orengo, Ph., Falourd, S., and Minster, B.: Rainfall and cave water isotopic relationships in two South-France sites, Geochimica et Cosmochimica Acta, 131, 323–343, https://doi.org/10.1016/j.gca.2014.01.043, 2014.

Hodell, D., Crowhurst, S., Skinner, L., Tzedakis, P. C., Margari, V., Channell, J. E. T., Kamenov, G., Maclachlan, S., and Rothwell, G.: Response of Iberian Margin sediments to orbital and suborbital forcing over the past 420 ka, Paleoceanography, 28, 185–199, https://doi.org/10.1002/palo.20017, 2013.

Martrat, B., Jimenez-Amat, P., Zahn, R., and Grimalt, J. O.: Similarities and dissimilarities between the last two deglaciations and interglaciations in the North Atlantic region, Quaternary Science Reviews, 99, 122–134, https://doi.org/10.1016/j.quascirev.2014.06.016, 2014.

Sinclair, D. J.: Two mathematical models of Mg and Sr partitioning into solution during incongruent calcite dissolution, Chemical Geology, 283, 119–133, https://doi.org/10.1016/j.chemgeo.2010.05.022, 2011.

Thomas, E. R., Wolff, E. W., Mulvaney, R., Steffensen, J. P., Johnsen, S. J., Arrowsmith, C., White, J. W. C., Vaughn, B., and Popp, T.: The 8.2ka event from Greenland ice cores, Quaternary Science Reviews, 26, 70–81, https://doi.org/10.1016/j.quascirev.2006.07.017, 2007.